# Tolerance and Metabolization of High-Concentration Heavy Crude Oil High-Concentration Heavy Crude Oil by *Bacillus subtilis*

**DOI:** 10.3390/microorganisms13071520

**Published:** 2025-06-29

**Authors:** César Antonio Sáez-Navarrete, Jessica Zerimar Cáceres-Zambrano

**Affiliations:** 1Departamento de Ingeniería Química y Bioprocesos, Pontificia Universidad Católica de Chile, Avenida Vicuña Mackenna 4860, Macul, Santiago 7820436, Chile; csaezn@uc.cl; 2Centro de Investigación en Nanotecnología y Materiales Avanzados (CIEN-UC), Pontificia Universidad Católica de Chile, Avenida Vicuña Mackenna 4860, Macul, Santiago 7820436, Chile

**Keywords:** heavy crude oil metabolization, bioupgrading, bioconversion, biotensioactives biotransformation capacity

## Abstract

In this comprehensive study, we investigated the degradation capacity and tolerance of the bacterial strain *Bacillus subtilis* in culture media with high concentrations of heavy crude oil (HCO) as the sole carbon source. Using a meticulously designed experimental approach conducted at room temperature (25 °C), we systematically examined various culture media with HCO concentrations of 20%, 35%, and 50% *v*/*v* over a 10-week period. The results revealed the microorganism’s remarkable resistance to these HCO concentrations. Biotransformation capacity was confirmed by quantifying CO_2_ production via gas chromatography, showing substantial bioconversion with a 42% increase in CO_2_ production. Additionally, changes in surface tension were monitored using the Du Noüy ring method, showing a reduction in the aqueous phase tension from 72.3 to 47.43 mN/m. At the end of the bioconversion period, all treated samples exhibited visible emulsification, indicative of biosurfactant production. This phenomenon was consistent with the observed decrease in surface tension, providing further evidence of biosurfactant-mediated mechanisms. These findings highlight the immense biotechnological potential of *B. subtilis* to address HCO-related challenges, offering promising prospects for crude oil bioremediation and bioupgrading.

## 1. Introduction

Biotechnology applied to petroleum upgrading offers a sustainable alternative for enhancing heavy crude oil (HCO), with the goal of reducing coke formation and greenhouse gas emissions by leveraging the metabolic capabilities of microorganisms. Among these strategies, ex situ bioupgrading relies on the microbial transformation of the heaviest and most polar fractions of crude oil, such as resins and asphaltenes, into lighter compounds. This process results in lower viscosity and increased API (American Petroleum Institute) gravity [1,2,3,4,5,6,7].

The microbial degradation of hydrocarbons generally follows an order based on increasing resistance to biodegradation: starting with alkanes, followed by branched alkanes, small aromatics, cyclic alkanes, polycyclic aromatics, resins, and finally asphaltenes [8]. Various strains from the genera *Bacillus* and *Pseudomonas* have been isolated from crude oil-contaminated environments [9]. For example, *Bacillus subtilis* and *Pseudomonas aeruginosa* strains isolated from oil-contaminated soil were used to extract lipase enzymes, which were then applied to crude oil degradation. The lipase from *B. subtilis* achieved a degradation rate of 8.11 ± 0.70%, while that from *P. aeruginosa* reached 15.69 ± 0.03% over a 28-day period. Another study evaluated a consortium of microbial strains (*Pseudomonas* sp., *Bacillus licheniformis*, *Bacillus lentus*, *Bacillus cereus*, and *Bacillus firmus*), which degraded between 35% and 48% of asphaltenes after 2 months of incubation at temperatures between 28 and 40 °C, using either pure or mixed cultures [10,11].

Historically, bioupgrading strategies have emerged from bioremediation research, which aims to stimulate microbial metabolism to transform or mineralize hydrocarbons in contaminated environments. However, bioremediation studies typically use low crude oil concentrations (1–4% *v*/*v*), which limits their applicability when evaluating microbial performance under the high-concentration conditions required for effective bioupgrading. Recent studies have begun to explore microbial activity and tolerance at higher concentrations of hydrocarbons. For instance, species of *Bacillus* have shown the ability to degrade asphaltenes at concentrations up to 20 g per liter, producing biosurfactants such as cyclic lipopeptides and surfactin-like compounds [8]. These biosurfactants are essential for increasing the bioavailability of hydrophobic compounds and for reducing the interfacial tension between the oil and water phases. Likewise, *P. aeruginosa* has demonstrated the capacity to grow in media containing 20% *v*/*v* of heavy vacuum gas oil (HVGO), although with limited influence on the distillation profile [12].

Although there is extensive literature on hydrocarbon-degrading bacteria, especially from the genera *Pseudomonas* and *Bacillus*, there is limited information on bacterial performance at very high concentrations of crude oil, specifically above 20% *v*/*v*, where oil is the sole carbon source. Most existing studies are focused on environmental remediation. In contrast, this study addresses a knowledge gap related to the metabolic behavior of *B. subtilis* under conditions that replicate bioupgrading applications instead of cleanup processes.

In this research, carbon dioxide production was used as an indicator of microbial metabolic activity and petroleum degradation. Under aerobic conditions, hydrocarbon mineralization results in carbon dioxide release, which provides a quantitative and non-destructive method for monitoring microbial respiration and substrate utilization [13,14,15]. While traditional indicators of biodegradation may be inconsistent or inadequate at high oil concentrations, carbon dioxide evolution enables continuous evaluation of microbial activity in heavy hydrocarbon environments.

In addition to this metabolic indicator, visual emulsification and oil droplet size analysis were carried out to gain further insights into the oil-in-water systems formed during the bioconversion process. Droplet dispersion and size distribution are strongly associated with biosurfactant activity and play a critical role in oil bioavailability and microbial access to hydrophobic substrates [16,17]. Furthermore, biosurfactant production was assessed through surface tension measurements, providing complementary evidence of microbial adaptation to the presence of heavy oil.

Although *B. subtilis* is recognized as a hydrocarbon-degrading and biosurfactant-producing bacterium [9,18], to our knowledge, no prior study has investigated its behavior in media containing 20, 35, and 50% *v*/*v* of heavy crude oil as the exclusive carbon source. Therefore, this study presents a novel approach to assess the potential of *B. subtilis* as a candidate for microbial upgrading of heavy crude oil.

The objectives of this study were the following:To identify the bacterial strain using molecular sequencing.To assess carbon dioxide production as an indirect metric of metabolic activity and petroleum degradation.To characterize emulsion formation and analyze oil droplet size as evidence of surfactant-mediated dispersion mechanisms.To evaluate biosurfactant production based on changes in surface tension.

## 2. Materials and Methods

The heavy crude oil used in this study was provided by Empresa Nacional del Petróleo de Chile (ENAP) (Santiago, Chile). This crude oil comes from Ecuador and has specific characteristics, including an API gravity of 17.1° ≅ 833 kg/m^3^, a viscosity of 348.48 cST at 50 °C, and a sulfur content of 2.411%. The bacteria used in this research were obtained from the Chilean Collection of Typical Strain Cultures (CCCT) of the Scientific and Technological Bioresources Nucleus BIOREN-UFRO (Temuco, Chile), attached to the “Universidad de La Frontera”. The bacterial strain is cataloged under the number CCCT 18.217 and was officially registered on 28 June 2018.

### 2.1. Preparation of Inoculum

The bacterial strains were maintained and cultured on Sabouraud agar, which consists of peptone (10 g/L), glucose or dextrose (20 g/L), and agar (15 g/L) [19]. Although Sabouraud agar is conventionally used for fungal cultivation due to its acidic pH and high sugar content, in this study, it was deliberately employed as a preconditioning medium. The slightly acidic and carbohydrate-rich environment imposed metabolic stress on *Bacillus subtilis*, which served to activate adaptive cellular responses. This strategy aimed to enhance the bacterium’s tolerance and metabolic flexibility prior to its exposure to heavy crude oil (HCO), a structurally complex and hydrophobic substrate significantly more challenging than simple carbohydrates like glucose. The use of Sabouraud agar was therefore intended to promote the selection of subpopulations more resistant to hostile conditions, such as those present in the subsequent bioconversion assays. The cultures were incubated under static conditions at ambient temperature (25–30 °C) for 48 h, a duration commonly used in pre-adaptation protocols for Gram-positive strains. This incubation time was selected to allow sufficient metabolic activation without overgrowth or nutrient depletion.

The inoculum was then prepared in potato dextrose broth (PDB), which was formulated using 200 g of potato, 20 g of dextrose, and 1000 mL of distilled water. The potato was boiled in 500 mL of distilled water for 30 min. The supernatant was then collected, and 20 g of dextrose was added. Subsequently, distilled water was added to adjust the final volume to 1000 mL. This broth was autoclaved for 20 min and, once cooled, was ready for bacterial culture [20].

### 2.2. Identification of Microorganisms

DNA extraction was initially performed using two commercial kits: one based on silica column purification and another based on salt precipitation specific for Gram-positive bacteria. Both approaches yielded insufficient DNA quantities for sequencing (0.124 ng/µL and 0.036 ng/µL, respectively), as measured by Qubit fluorometry.

Given the low DNA recovery and the age of the bacterial sample (originally obtained in 2018 from a certified strain collection), a third attempt was made using a soil DNA kit with modified lysis conditions, including ultrasonic treatment and heating, but this also failed to produce a usable sequence.

As a final approach, the E.Z.N.A.^®^ Forensic DNA Kit (Omega Bio-Tek) (Norcross, GA, USA) was used due to its suitability for degraded or recalcitrant samples. The protocol was enhanced by including ultrasonic incubation at 80 °C for 90 min prior to DNA binding. This method successfully yielded 90.2 ng/µL of DNA, which enabled PCR amplification using primers 27F and 1492R. In parallel, identification was confirmed using MALDI-TOF/TOF (matrix-assisted desorption/ionization time-of-flight) analysis (Autoflex Speed, Bruker) (Bremen, Germany). A P-GEM vector was used as a sequencing control, and identity was verified using BLAST (Sequencing Analysis v6.0 and Geneious Prime^®^ version 2022.2.2 were used for sequence analysis and alignment).

### 2.3. Biotransformation Test

Three Czapeck liquid nutrient media were prepared, devoid of agar, and consisting of the following components: dipotassium phosphate (1 g/L), a concentrated salt solution (10 mL/L), yeast extract (5 g/L), and sucrose (30 g/L).

The concentrated salt solution, a key component, was composed of distilled water (100 mL), sodium nitrate (30 g), potassium chloride (5 g), magnesium sulfate heptahydrate (5 g), and ferrous sulfate. This solution was sterilized in an autoclave at 120 °C for 15 min.

These media were formulated by replacing the carbon source (sucrose) with heavy crude oil at concentrations of 20% *v*/*v*, 35% *v*/*v*, and 50% *v*/*v*. The solution was stored at room temperature in a sealed bottle [21].

Each nutrient solution was dispensed into 300 mL bottles at a volume of 200 mL. In addition to the experimental media, a positive control consisting of Czapeck liquid nutrient medium with yeast extract and sucrose was prepared and inoculated with bacteria. A negative control, consisting of a culture medium containing 35% *v*/*v* uninoculated heavy crude oil, was also established. These media were prepared in triplicate, resulting in a total of 15 bottles.

All bottles were stored at room temperature (25–30 °C) and subjected to orbital shaking at 200 rpm for 10 weeks [22].

### 2.4. CO_2_ Production

Headspace analysis was conducted daily during the first week, followed by measurements every other day for the next 2 weeks. Subsequently, weekly measurements were performed for a total duration of 10 weeks. For each analysis, 250 µL of the sample was extracted and analyzed using an Agilent Technologies 7820A gas chromatograph. This instrument was equipped with an Agilent 250361-01 Carboxen 1010 Plot column and a thermal conductivity detector (TCD) (Santa Clara, CA, USA). The oven temperature was maintained at 200 °C, while the detector temperature was set to 230 °C. The operating temperature range was −60 °C to 250 °C, and the column dimensions were 30 m × 530 µm × 30 µm.

To quantify the results, a calibration curve was previously established to correlate CO_2_ concentration with peak area. The data obtained from the analysis are expressed as a percentage of the total volume generated over the weeks [22].

### 2.5. Droplet Size Analysis

At the end of the incubation period, the culture media containing heavy crude oil and microorganisms exhibited clear emulsification, with visible formation of dispersed oil droplets throughout the system. To facilitate visualization and analysis, a portion of each sample was carefully transferred to 250 mL Erlenmeyer flasks with a known base diameter of 75 mm, allowing for clearer observation of the emulsions. The samples were photographed directly in the flasks, without agitation, using natural lighting and a fixed camera distance.

The resulting images were used to perform droplet size analysis through digital segmentation. The scale was calibrated based on the known diameter of the flask (75 mm = 300 pixels), establishing a conversion factor of 4 pixels per millimeter. Image processing and droplet analysis were carried out using ImageJ software (version 1.53t), including grayscale conversion, thresholding, and contour detection for identifying individual droplets. The droplet areas obtained in pixels^2^ were converted into real surface units (mm^2^) using the relationship (1 mm^2^ = 16 pixels^2^) [16]. This procedure enabled the quantification of the average size and distribution of droplets present in the emulsions formed after bioconversion

### 2.6. Changes in Surface Tension

Sample measurements were conducted at the same frequency as the headspace analysis using a Krüss tensiometer, model Easy Dyne K20, manufactured by Krüss in Hamburg, Germany. This instrument was equipped with a platinum ring and operated at room temperature, maintained between 25 °C and 30 °C. Measurements were performed following the Du Noüy ring method.

To ensure accuracy, the instrument was calibrated by adjusting the measurements to achieve a water surface tension reading of 72 mN/m. The culture media were then centrifuged at 5000 rpm for 10 min [22].

## 3. Results

### 3.1. Identification of Microorganisms

The microbial samples were analyzed using the MALDI-TOF/TOF Autoflex Speed system, manufactured by Bruker Daltonics GmbH, Bremen, Germany (Table 1).

Additionally, bioinformatics analysis (including cleaning of Sequencing Analysis v6.0 reads, assembly of paired reads with Geneious Prime^®^ 2022.2.2, and obtaining a consensus sequence) confirmed *B. subtilis* with 100% coverage and 99.9% identity. The consensus sequence is 1329 bp long (see Figure 1).

The P-GEM vector was used as a sequence control, and a BLAST analysis resulted in 100% coverage and 100% identity (see Figure 2).

### 3.2. Metabolization and Tolerance to Heavy Crude Oil

CO_2_ production was monitored using headspace analysis via gas chromatography. Figure 3 displays the CO_2_ production over time. In the media where heavy crude oil (HCO) was the sole carbon source, the percentage by volume of CO_2_ increased progressively until days 30 and 37, while the positive control (sucrose medium) reached 60% CO_2_ by day 7. The negative control (without inoculum) showed no significant CO_2_ production.

To evaluate the significance of the observed differences in CO_2_ production, a two-way ANOVA was performed, considering time and treatment as factors. The results, presented in Table 2, show that the “sample” factor (corresponding to treatment groups and time) was statistically significant (*p* < 0.001). This indicates that CO_2_ production varies significantly depending on the treatment and time, supporting the metabolic activity of *B. subtilis* in HCO media.

### 3.3. Biosurfactant Production

After the bioconversion period, agitation was stopped; however, no phase separation was observed. Instead, emulsions in the creaming stage were present, as shown in Figure 4, where oil droplets remained visibly suspended and separated from each other within the aqueous medium. This behavior indicates the presence of surface-active compounds, likely biosurfactants, produced during the process.

Images of the emulsions were captured and processed using ImageJ software for droplet size analysis (Figure 5). The quantitative results are summarized in Table 3 and represented graphically in Figure 6. The average droplet diameters for media containing 20% and 35% heavy crude oil (HCO) were very similar. In contrast, in the medium containing 50% HCO, the droplets were substantially larger.

As an exploratory step, part of a sample was passed through a filter to assess whether the oil droplets would easily coalesce. However, the droplets remained separated, further suggesting the presence of stabilizing agents that inhibit coalescence (Figure 7).

Surface tension was measured to monitor biosurfactant production. Figure 8 shows the evolution of surface tension over time. Control measurements were performed using Milli-Q water (72 mN/m) and HCO alone (30 mN/m). The negative blank (culture medium with 35% HCO without inoculum) recorded 30.80 mN/m. In contrast, the inoculated media exhibited fluctuations in surface tension, with the most substantial change observed on day 7, especially in the 20% HCO medium.

A statistical analysis (two-way ANOVA) was performed on the triplicate data to evaluate differences across conditions and over time. The analysis revealed significant differences between experimental groups (F = 14.70, *p* < 0.001), as well as a significant effect over time (F = 6.73, *p* < 0.001). No significant interaction was found (F = 0.158, *p* = 1), suggesting that the treatment effects remained consistent throughout the evaluation period. These findings support the observed stabilization of surface tension after day 15, particularly in the BCZ20 condition. Full ANOVA results are presented in Table 4. After the 10-week bioconversion process, the media were centrifuged (5000 rpm for 10 min), and the surface tension of the aqueous phase was measured. Figure 9 and Table 5 indicate that the 20% HCO medium experienced the most significant decrease in surface tension (47.43 mN/m).

## 4. Discussion

The results obtained provide several relevant insights into both the microbial identification and the biodegradation process of heavy crude oil (HCO). The confirmation of *B. subtilis* through MALDI-TOF/TOF analysis, supported by bioinformatics tools, reinforces the reliability of the identification method. Although the strain was obtained from a recognized strain collection and cataloged in 2018, the time elapsed justified confirmatory sequencing to ensure its authenticity. The identification was consistent with the aerobic nature of *B. subtilis*, which aligns with the experimental conditions and the observed carbon dioxide evolution.

Regarding substrate degradation, the progressive increase in carbon dioxide production in the HCO media indicates effective mineralization of the carbon source. The higher CO_2_ levels detected in the 20% and 35% HCO media suggest that these concentrations provided favorable conditions for microbial activity. In contrast, the reduced CO_2_ production observed at 50% HCO likely reflects substrate oversaturation, which may have hindered efficient metabolic processes (Figure 3). These findings are in agreement with previous studies [22,23], which also reported increased carbon dioxide evolution during the degradation of complex carbon compounds.

To better understand the physical limitations potentially influencing biodegradation at high hydrocarbon loads, a droplet size analysis was conducted using digital image segmentation of emulsified samples. The results (Figure 6, Table 3) revealed that droplet diameters were similar in the 20% and 35% HCO media; however, substantially larger droplets were observed in the 50% HCO condition. This increase in droplet size suggests reduced emulsification efficiency or diminished biosurfactant production at higher oil concentrations. Larger droplets have a lower surface-area-to-volume ratio, reducing microbial access to hydrocarbons and potentially limiting biodegradation rates [16,17,24].

These morphological observations support the CO_2_ data, indicating that the decreased mineralization at 50% HCO may be attributed not only to substrate oversaturation but also to limited interfacial contact and physical accessibility of the substrate. Additionally, the high viscosity and hydrophobicity of the medium at this concentration may have impaired oxygen transfer and nutrient diffusion, contributing to cellular stress and metabolic suppression [25,26]. This is consistent with findings that excessive hydrocarbon content can inhibit microbial growth and respiration due to toxic effects or physical constraints [27].

The formation of emulsions in the creaming stage (Figure 4) further illustrates the emulsification dynamics. After incubation, no clear phase separation occurred, and droplets were visually suspended within the aqueous phase. Experimental filtering showed that the droplets remained separated, resisting coalescence (Figure 7), suggesting the presence of surface-active microbial products. The emulsified systems were stable enough to permit image-based quantification, reinforcing the hypothesis of biosurfactant-mediated stabilization.

The role of biosurfactant production is further elucidated by the surface tension measurements. It is important to highlight that Figure 8 presents surface tension values obtained directly from the complete culture medium, without phase separation. Therefore, these measurements reflect a complex matrix composed of both heavy oil and extracellular microbial products. Within this mixture, an initial increase in surface tension was observed, particularly during the first week of the bioconversion process. This behavior can be attributed to early interactions between emerging biosurfactants and heavy hydrocarbon chains, possibly leading to the formation of emulsified structures or aggregates that temporarily raise interfacial tension. Similar effects have been described in previous studies [28], which demonstrated that interactions between surfactant head groups and oil molecules can result in variable interfacial tension responses, depending on the chemical environment and surfactant concentration.

To better assess biosurfactant activity, surface tension was also measured in the aqueous phase after centrifugation at the end of the bioconversion period. In this clarified phase, a substantial decrease in surface tension was recorded, confirming the presence of extracellular biosurfactants produced by *B. subtilis* (Figure 9). This reduction strongly indicates the production of amphiphilic microbial compounds that enhance the solubility and availability of hydrophobic substrates, thereby promoting more effective biodegradation [29,30,31]. The marked change in surface tension, especially during the first 7 days in the 20% HCO medium, supports the onset of biosurfactant production. Similar trends were observed in previous studies [18,32,33], where biosurfactant synthesis during the exponential phase was associated with a significant reduction in surface tension.

The statistical significance of the results (*p* < 0.05) further supports the reliability and consistency of these observations. Overall, the findings demonstrate that *B. subtilis* is capable of adapting to and metabolizing high concentrations of heavy crude oil while producing compounds with surfactant properties that improve substrate accessibility. The combination of validated microbial identification, consistent trends in carbon dioxide evolution, measurable reductions in surface tension, and quantitative droplet size analysis provides a comprehensive understanding of the biodegradation process. These results indicate that *B. subtilis* holds considerable potential as a biological upgrading agent for heavy crude oil. These outcomes are also in line with previous results obtained for *Aspergillus flavus* [22]. Moreover, the experimental design used in this study, which employed crude oil concentrations above 20% (*v*/*v*) as the sole carbon source, presents a novel approach to evaluating the biodegradation capacity of *B. subtilis* beyond traditional bioremediation applications.

## 5. Study Limitations

While this study provides compelling evidence of *B. subtilis* activity in the bioconversion of heavy crude oil (HCO) at high concentrations, several limitations should be acknowledged. First, the experiments were conducted under strictly controlled laboratory conditions, which do not fully replicate the environmental complexity and variability of real-world settings. Factors such as microbial competition, oxygen availability, and dynamic fluid properties were not assessed and may influence biodegradation performance in situ.

Second, the study did not include chemical analysis of intermediate metabolites (e.g., organic acids, aliphatic/aromatic degradation products) or compositional profiling of crude oil before and after treatment. This was due to limited access to analytical instrumentation and reagents. However, the combination of CO_2_ evolution, surface tension measurements, and droplet size analysis offers strong and converging lines of evidence to support the metabolic activity and biosurfactant production capabilities of *B. subtilis*.

Moreover, the duration of the study (10 weeks) may not capture long-term emulsion stability or complete hydrocarbon mineralization, particularly under substrate saturation conditions. Nonetheless, the results provide a robust proof-of-concept that highlights the feasibility of using *B. subtilis* for high-load bioconversion of HCO and lays a solid foundation for future research on oil upgrading through microbial means.

## 6. Future Directions

Based on the findings and limitations of this study, several future research directions are proposed to deepen the understanding and promote the practical application of *B. subtilis* in the bioconversion of heavy crude oil.

First, a detailed compositional analysis of crude oil before and after microbial treatment is recommended. This would help identify chemical changes that occur during biodegradation, including the presence of intermediate metabolites such as organic acids or products resulting from the breakdown of aliphatic and aromatic compounds. This information would provide valuable insights into the underlying metabolic pathways.

Second, longer-term studies are needed to evaluate the stability of emulsions, the persistence of biosurfactant activity, and the extent of hydrocarbon mineralization over time. Conducting experiments in bioreactors under conditions that simulate continuous operation would allow for a more realistic assessment of potential industrial applications.

Third, the optimization of culture conditions should be explored to enhance process efficiency. Factors such as nutrient availability, pH regulation, temperature, and oxygen levels can significantly influence microbial growth and biosurfactant production. Additionally, incorporating co-cultures with other bacteria or oil-tolerant fungi may lead to synergistic effects that improve overall performance.

Finally, future research should include economic and environmental evaluations such as feasibility studies and life cycle assessments. These approaches will be essential for determining the potential for scaling up *B. subtilis*-based systems for the biological upgrading of crude oil. Integrating microbiological, engineering, and economic knowledge will be key to developing sustainable and practical industrial solutions.

## 7. Conclusions

This study demonstrates that *B. subtilis* is capable of surviving and metabolizing heavy crude oil (HCO) at high concentrations (20, 35, and 50% *v*/*v*), using it as the sole carbon and energy source. CO_2_ evolution profiles confirmed aerobic metabolic activity, particularly during the first 15 days of incubation, even under the most challenging condition (50% *v*/*v*). Notably, *B. subtilis* showed the highest metabolic performance in the 20% and 35% HCO treatments, with cumulative CO_2_ production increases of 42% and 39%, respectively, compared to the control.

The observed reduction in surface tension across all treatments further supports the production of biosurfactants, which likely facilitated hydrocarbon uptake and microbial adaptation to the hydrophobic environment. In the 20% HCO condition, surface tension decreased significantly from 72.3 to 47.43 mN/m, suggesting effective biosurfactant secretion and enhanced oil emulsification.

Moreover, droplet size analysis revealed key insights into the physical behavior of the emulsified systems. While emulsions in the 20% and 35% HCO conditions exhibited relatively uniform and small droplets, the 50% HCO treatment displayed significantly larger droplets, indicating reduced emulsification efficiency. This morphological evidence correlates with the lower CO_2_ production in the 50% condition and suggests that physical substrate accessibility may also limit metabolic performance at high oil loads.

These findings provide the first experimental evidence of *B. subtilis* activity under such elevated crude oil concentrations, reinforcing its potential as a microbial agent for ex situ bioupgrading applications. The results highlight the dual capability of *B. subtilis* to tolerate petroleum toxicity and to enhance hydrocarbon bioavailability through biosurfactant production and emulsion stabilization.

Further research should explore the compositional changes in crude oil after treatment, as well as optimization of culture conditions (e.g., nutrient amendments, co-cultures, reactor configurations) to enhance bioconversion efficiency. Integrating *B. subtilis* into fungal or bacterial consortia, or into biorefinery strategies, could contribute to more sustainable and efficient upgrading of heavy petroleum fractions.

## Figures and Tables

**Figure 1 microorganisms-13-01520-f001:**
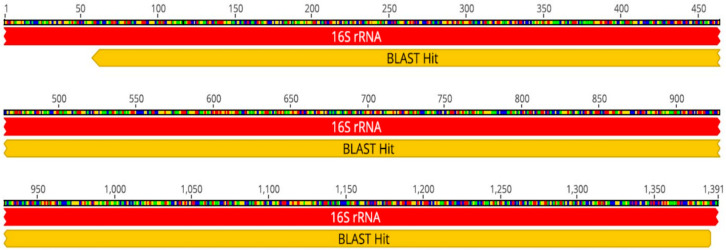
Blast sequence of *Bacillus subtilis* sample consensus.

**Figure 2 microorganisms-13-01520-f002:**
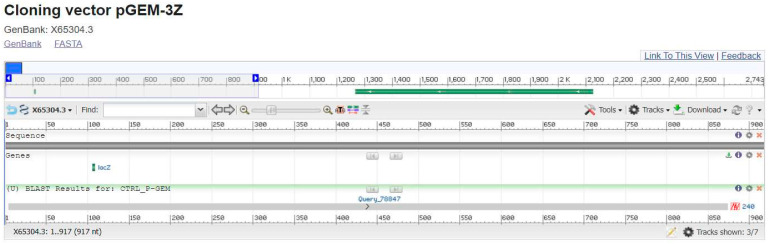
Blast shows CTRL P-GEM.

**Figure 3 microorganisms-13-01520-f003:**
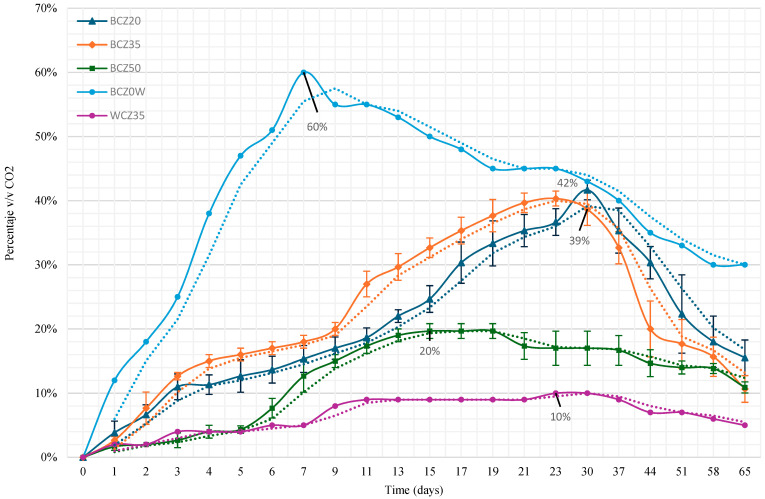
CO_2_ production according to culture medium: BCZ20 (Czapeck medium with yeast extract inoculated with bacteria and 20% HCO), BCZ35 (with 35% HCO), BCZ50 (with 50% HCO), BCZ0W (positive control: sucrose medium with bacteria), and WCZ35 (negative control: 35% HCO without inoculum). Data represent the mean values of triplicate samples, and error bars indicate the corresponding standard errors.

**Figure 4 microorganisms-13-01520-f004:**
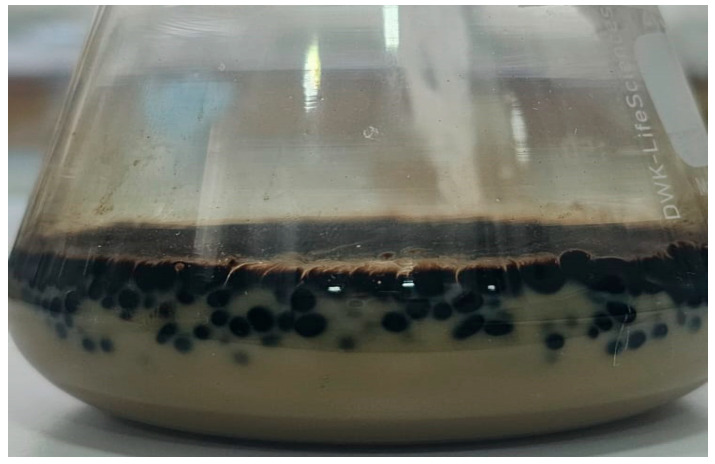
Image of one of the samples where the emulsion can be clearly observed in the creaming phase.

**Figure 5 microorganisms-13-01520-f005:**
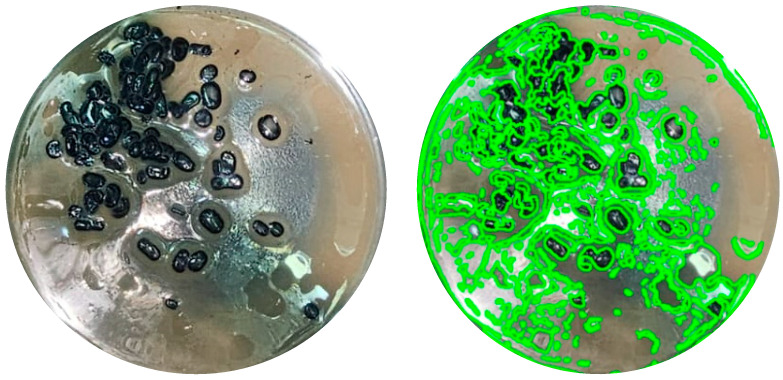
Image capture of one of the samples processed in ImageJ, showing pixel identification and segmentation of droplets as a preliminary step for droplet size calculation.

**Figure 6 microorganisms-13-01520-f006:**
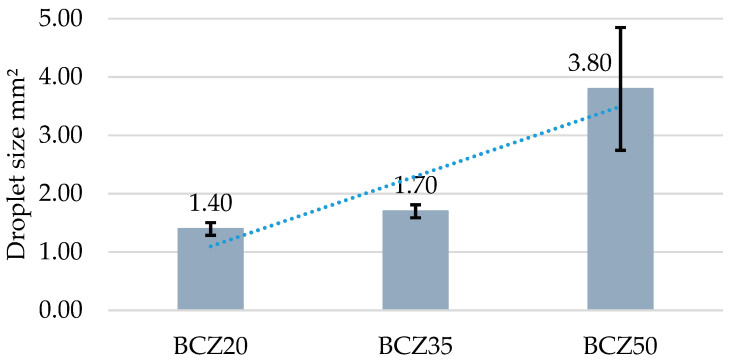
Droplet diameter distribution according to each culture medium. BCZ20 corresponds to medium containing 20% *v*/*v* of heavy crude oil (HCO), BCZ35 to 35% *v*/*v* HCO, and BCZ50 to 50% *v*/*v* HCO.

**Figure 7 microorganisms-13-01520-f007:**
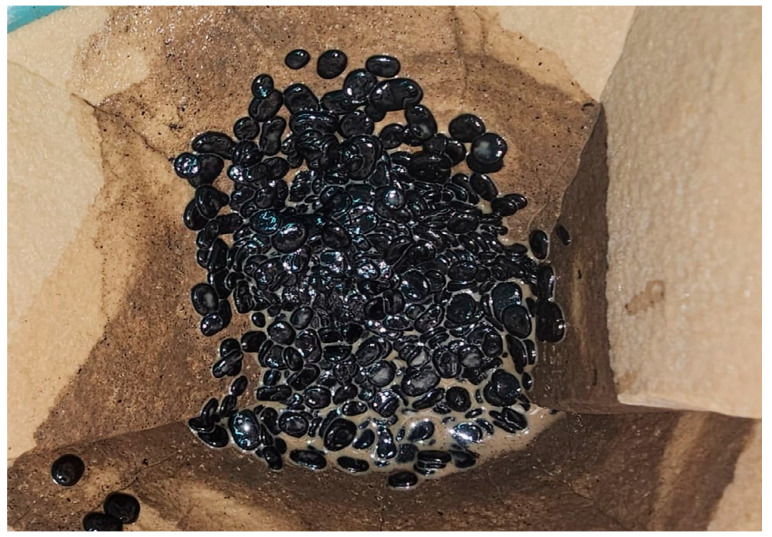
Filtered sample showing dispersed oil droplets that remained separated.

**Figure 8 microorganisms-13-01520-f008:**
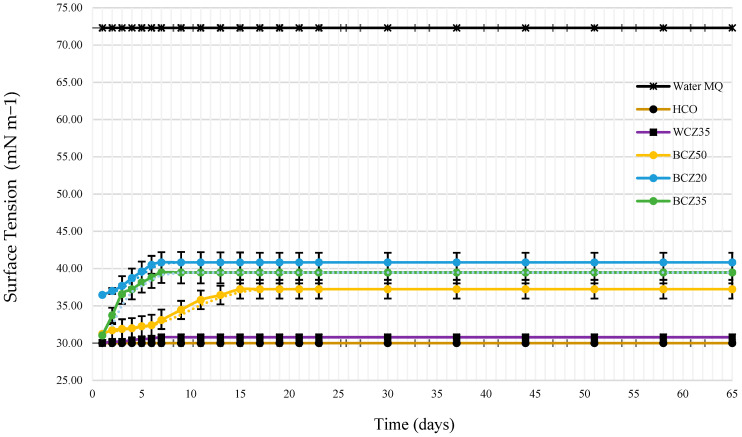
The surface tension of the O/W emulsion was determined according to the following culture media: heavy crude oil HCO (control), WCZ35 Czapeck medium with yeast extract and 35% HCO without inoculum (control), BCZ20 Czapeck medium with yeast extract inoculated with *B. subtilis* and 20% HCO, BCZ35 Czapeck medium with yeast extract inoculated with *B. subtilis* and 35% HCO, BCZ50 Czapeck medium with yeast extract inoculated with *B. subtilis* and 50% HCO.

**Figure 9 microorganisms-13-01520-f009:**
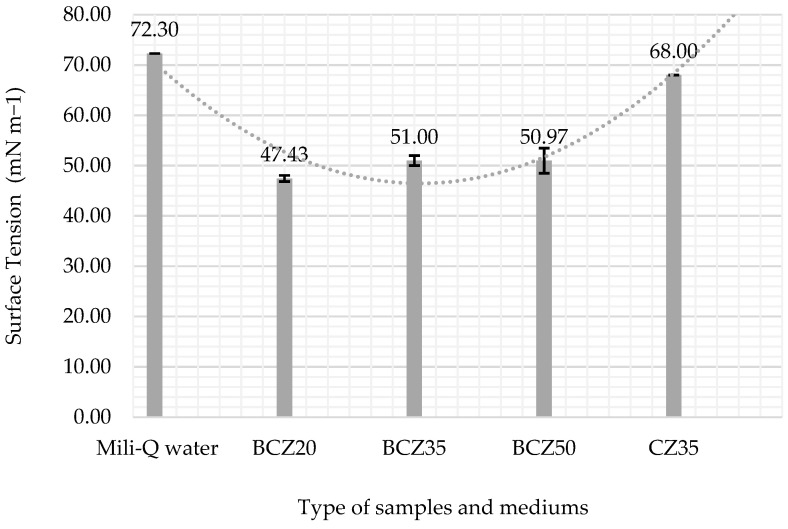
Changes in the surface tension of the aqueous phase: included control Mili-Q water, the CZ35 aqueous phase of medium with 35% HCO without inoculum (control), the BCZ20 aqueous phase of inoculated medium with 20% HCO, the BCZ35 aqueous phase of inoculated medium with 35% HCO, and the BCZ50 aqueous phase of inoculated medium and 50% HCO. The data are the mean values of triplicate samples. The error bars represent the corresponding standard errors.

**Table 1 microorganisms-13-01520-t001:** Results using the MALDI-TOF/TOF Autoflex Speed system.

Code	Log (Score)	Identification
1	2.220	*Bacillus subtilis* ssp. *subtilis*
2	2.190	*Bacillus subtilis* ssp. *subtilis*
3	2.170	*Bacillus subtilis* ssp. *subtilis*

**Table 2 microorganisms-13-01520-t002:** Two-way ANOVA for CO_2_ production across different media containing heavy crude oil (HCO) over time.

ANOVA						
Source of Variation	SS	df	MS	F	*p*-Value	F Crit
Sample	1.804439	21	0.085926	16.73094	1.12 × 10^−27^	1.63655
Columns	0.000507	2	0.000254	0.049361	0.951855	3.064761
Interaction	0.022684	42	0.00054	0.105163	1	1.477077
Within	0.677917	132	0.005136			
Total	2.505547	197				

**Table 3 microorganisms-13-01520-t003:** Descriptive statistics per sample group (in mm^2^).

Sample	Mean Area (mm^2^)	Standard Deviation	Minimum (mm^2^)	Maximum (mm^2^)	n
BCZ20	1.396	0.110	1.281	1.500	3
BCZ35	1.698	0.110	1.594	1.813	3
BCZ50	3.796	1.054	2.594	4.563	3

**Table 4 microorganisms-13-01520-t004:** Two-way ANOVA for changes in surface tension in different media containing heavy crude oil (HCO) over time.

ANOVA						
Source of Variation	SS	df	MS	F	*p*-Value	F Crit
Sample	1.3047	24	0.07412	14.701	1.28 × 10^−9^	2.45576
Columns	0.0033	2	0.00367	6.7293	2.76 × 10^−13^	1.58927
Interaction	0.0627	48	0.00065	0.1581	1	1.87371
Within	0.7787	150	0.00346			
Total	2.1494	224				

**Table 5 microorganisms-13-01520-t005:** Descriptive statistics per sample group (in mm^2^).

Sample	Surface Tension (mN m^−1^)	Standard Deviation	Minimum (mN m^−1^)	Maximum (mN m^−1)^	n
Mili-Q water	72.3	0.1	72.2	72.3	3
BCZ20	47.43	0.60	46.8	48.1	3
BCZ35	51.00	1	50	52	3
BCZ50	50.97	2.5	49.76	51	3
CZ35	68	0.2	67.9	68.1	3

## Data Availability

The original contributions presented in the study are included in the article, further inquiries can be directed to the corresponding author.

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
