# Peer review of "Tolerance and Metabolization of High-Concentration Heavy Crude Oil High-Concentration Heavy Crude Oil by Bacillus subtilis"

_microorganisms, 2025, doi:10.3390/microorganisms13071520_

Round 1
Reviewer 1 Report (New Reviewer)
Comments and Suggestions for Authors
This study is innovative, but it needs to be modified and the terminology and graphs standardized.
Specific comments:
1. The negative control chose a medium containing 35% volume ratio without inoculation of heavy crude oil because there was a reference or a pre-experiment
2. Studies have confirmed the production of biosurfactants, but no specific compounds (e.g., surfactin) have been identified. Supplemental FTIR or HPLC data, if available, is recommended.
Are intermediate metabolites (e.g., organic acids, alkane/aromatic degradation products) analyzed for 3.COâ‚‚ generation as a degradation indicator? If not, it needs to be stated in the limitations.
4. Analyze whether biosurfactants are directly related to crude oil emulsification (supplemental microscopic observation or droplet size analysis is recommended)
5. Discuss the reason for the lower CO2 production in the 50% HCO group, whether it is due to substrate inhibition or cellular stress?
6. It is necessary to explain the changes in the composition of the crude oil that have not been analyzed (e.g., asphaltene reduction, API gravity change)
7. Abbreviations such as HCO, API, MALDI-TOF/TOF should be given full names when they appear for the first time
8. Application potential: Inhibition of activity at 50% HCO suggests that conditions need to be optimized (e.g., nutritional supplementation or microbiotic symbiosis).
Author Response
I sincerely thank you for the time and dedication invested in reviewing our manuscript, as well as for your valuable comments and suggestions. Your feedback has been instrumental in strengthening the scientific quality of the work.
In the attached document, you will find a detailed response addressing each of the points you raised. We have made the corresponding modifications to the manuscript and hope that the changes will be satisfactory.

Reviewer 2 Report (Previous Reviewer 2)
Comments and Suggestions for Authors
This study presents valuable insights into B. subtilis’ potential for high-concentration HCO bioupgrading. With methodological clarifications, better statistical analysis, and expanded discussion, it could be a good contribution to Microorganisms.
Title: „subtilis” should be written in lowercase.
State why Sabouraud agar (typically used for fungi) was utilized for a bacterium. Specify the incubation conditions, such as temperature and duration.
Clarify the rationale behind applying forensic DNA protocols to bacterial samples.
Figure 3.4 – Why is there a mention about fungal inoculation?
The authors perform statistical analysis only for one figure.
Author Response
Les agradezco sinceramente el tiempo y la dedicación invertidos en la revisión de nuestro manuscrito, así como sus valiosos comentarios y sugerencias. Sus comentarios han sido fundamentales para fortalecer la calidad científica del trabajo.
En el documento adjunto encontrará una respuesta detallada que aborda cada uno de los puntos planteados. Hemos realizado las modificaciones correspondientes al manuscrito y esperamos que los cambios sean satisfactorios.

Round 2
Reviewer 2 Report (Previous Reviewer 2)
Comments and Suggestions for Authors
The authors addressed all the comments and carefully revised the manuscript.
This manuscript is a resubmission of an earlier submission. The following is a list of the peer review reports and author responses from that submission.
Round 1
Reviewer 1 Report
Comments and Suggestions for Authors
Dear authors,
After reading your article, I have a number of questions and comments.:
- Lines 2-3 “Tolerance and metabolization of high-concentration heavy crude oil BY BACILLUS SUBTILIS” – please correct the font style and make it uniform.
- Line 88 – “(Pengnoo et al., 2005)”– references in the text should be added in the format approved by the rules of the journal, in square brackets.
- Line 114, Line 126, Line 135 – “(Cáceres-Zambrano et al., 2024)” – references in the text should be added in the format approved by the rules of the journal, in square brackets.
- Line 37 - [20; 11; 21; 22; 4; 3; 8] – The references in the text should be placed in order, starting with the number [1]. However, the number 1 in the manuscript is only on line 44 at the beginning of the line - “[1] An isolated Bacillus subtilis…”, while the reference number is usually placed at the end of the phrase. Please pay due attention to the format and order of the references in the text.
- Line 75 – “1°” replace with “17.1 °C”.
- Why the first figure is numbered Figure 3.1? The same applies to tables, the first table in the text is numbered - “Table 3. 1 Results MALDI-TOF/TOF Autoflex Speed”- Please correct the numbering of all figures and tables.
- What is new about your research? To date, a large number of articles have been published on the use of various bacterial strains, including the genus Bacillus, to study the biodegradation of oil and petroleum products. These articles detail both the efficiency of pollutant removal and the various genetic, biochemical and physiological characteristics of hydrocarbon-degrading bacteria.
- Section Introduction – Why do you mention microorganisms such as Pseudomonas aeruginosa and Aspergillus flavus if you use only a strain of the genus Bacillus in your research?
- Line 65 – “Based on this background, this study selected the bacterium subtilis…” – In my opinion, this justification for selecting a microorganism for research is not correct. The efficiency of oil utilisation can vary between strains within the same species, and it is not correct to say that in some previously published work a strain of one genus was better than a strain of another genus, and therefore you chose a representative of that genus. At the very least, you should have statistics on such work where, for example, bacteria of the genus Bacillus are more effective than bacteria of the genus Pseudomonas.
- Figure 3. – The legend for the drawing does not specify all types of lines located on the graph (in particular, these are blue and purple lines).
- For each figure and each table, a transcript of abbreviations should be added to the caption.
12. The introduction contains insufficient information about the relevance and novelty of the work. The purpose of the article should be the only one, and the identification of bacteria cannot be the purpose of scientific work, it also refers to the control of carbon dioxide and the assessment of the ability to produce biosurfactants. It is not clear why this study was conducted, to search for effective oil destructors, or to search for producers of biosurfactants to reduce the viscosity of oil in order to use these bacteria in the process of secondary oil production? There is a big question about setting up an experiment where the authors introduced oil in an amount of more than 20% v/V. The method of identification of the microorganism and the confident statement that the strain under study belongs to the species B. subtilis are also questionable. In the Materials and Methods section, and in the manuscript as a whole, there is no description of the method for identifying bacteria by the sequence of the 16S rRNA gene. Even if the authors intended this method, it is known that using the 16S rRNA gene as the only criterion for phylogenetic distance is problematic, especially for close taxa. For example, the sequences of the 16S rRNA gene may be identical for different Bacillus species. There is no drawing showing the phylogenetic tree of the identified microorganism. There is no evidence in the article that a microorganism can use petroleum hydrocarbons as the only source of carbon and energy, for example, growth curves of a bacterial strain in an environment with oil, chemical analysis of oil loss. When describing a newly isolated destructive strain, it is necessary to provide data on the physiology of the microorganism and its phenotypic features. Specify the source from which the microorganism was isolated. As for the analysis of surface tension, the results are questionable, since it is difficult to imagine how to measure the surface tension of a liquid culture medium with oil in the amount of 50% v/V. It is not clear why the surface tension increases with time (Figure 3.4)? However, in the Discussion section, you write that the surface tension was decreasing.
Reviewer 2 Report
Comments and Suggestions for Authors
The article is relatively weak, as the authors only investigated gas production and observed a decrease in surface tension. It lacks any statistical analyses, which significantly undermines the reliability of the results. The study of biosurfactants could be greatly improved by incorporating several simple methods based on spectrophotometric or other analytical techniques.
Line 76: Clarify whether the bacterial strain(s) obtained from the Chilean Collection of Typical Strain Cultures (CCCT) have assigned catalog/accession numbers. What is puzzling is the fact that you are undertaking the sequencing of strains that already belong to the culture collection. Table 3: The data suggest that all three samples analyzed were identified as B. subtilis. Please clarify whether this lack of diversity was expected or merits further discussion.
CO2 Production: The introduction does not adequately justify the focus on CO2 as a key metric. Expand the background to explain its relevance to microbial metabolism, environmental interactions, or the study’s specific hypotheses. Statistical Analysis: Provide details on statistical methods used (e.g., tests, significance thresholds) for quantitative results in Table 3.2 and similar datasets.
Discussion: The current section is underdeveloped. It should definitely be deepened and broadened.
Renumber figures and tables to follow their order of appearance in the text. Moreover, in the case of Table 3.2 and Figure 3.3: These appear to duplicate the same dataset. Consolidate into a single format (table or figure).
Lines 2–3: Ensure font consistency throughout the manuscript (e.g., lower- and uppercase letters).
Line 15: Latin names (e.g., genus and species) should be italicized
References: Please number references sequentially in the order they first appear in the text. Formatting should strictly adhere to the journal’s Instructions for Authors.
Line 49: The full scientific name of the bacterial species (e.g., Bacillus subtilis) should be spelled out in full upon first mention, followed by the abbreviated form (e.g., B. subtilis) in subsequent uses. Currently, this is inconsistently applied.
Lines 145–146: Remove this sentence.